# *Staphylococcus aureus* and Cystic Fibrosis—A Close Relationship. What Can We Learn from Sequencing Studies?

**DOI:** 10.3390/pathogens10091177

**Published:** 2021-09-13

**Authors:** Christine Rumpf, Jonas Lange, Bianca Schwartbeck, Barbara C. Kahl

**Affiliations:** Institute of Medical Microbiology, University Hospital Münster, 48149 Münster, Germany; Christine.Rumpf@ukmuenster.de (C.R.); jonaslange@yahoo.de (J.L.); Biancawa@uni-muenster.de (B.S.)

**Keywords:** *Staphylococcus aureus*, cystic fibrosis, whole genome sequencing, *spa* sequence typing, populations dynamics, persistent infection, antibiotic resistance, pathogenesis

## Abstract

*Staphylococcus aureus* is next to *Pseudomonas aeruginosa* the most isolated pathogen from the airways of cystic fibrosis (CF) patients, who are often infected by a dominant *S. aureus* clone for extended periods. To be able to persist, the pathogen has to adapt to the hostile niche of the airways to counteract host defence, antibiotic therapy and the competition with coinfecting pathogens. *S. aureus* is equipped with many virulence factors including adhesins, toxins that are localized on the chromosome, on plasmids or are phage-related. *S. aureus* is especially versatile and adaptation and evolution of the pathogen occurs by the acquisition of new genes by horizontal gene transfer (HGT), changes in nucleotides (single nucleotide variations, SNVs) that can cause a selective advantage for the bacteria and become fixed in subpopulations. Methicillin-resistant *S. aureus* are a special threat to CF patients due to the more severe lung disease occurring in infected patients. Today, with decreasing costs for sequencing, more and more studies using *S. aureus* isolates cultured from CF patients are being published, which use whole genome sequencing (WGS), multilocus sequence typing (MLST) or *spa*-sequence typing (*spa*-typing) to follow the population dynamics of *S. aureus*, elucidate the underlying mechanisms of phenotypic variants, newly acquired resistance or adaptation to the host response in this particular niche. In the first part of this review, an introduction to the genetic make-up and the pathogenesis of *S. aureus* with respect to CF is provided. The second part presents an overview of recent studies and their findings using genotypic methods such as single or multilocus sequencing and whole genome sequencing, which identify factors contributing to the adaptation of *S. aureus* and its evolution in the airways of individuals with CF.

## 1. Pathogenesis of *S. aureus* with Respect to Cystic Fibrosis

The genetic disease cystic fibrosis (CF) is caused by mutations in an important chloride channel called the CF transmembrane conductance regulator (CFTR). The resulting dysfunction alters the ion transport of cell membranes [1]. Therefore, viscous mucus is formed in many organs, including the lungs, making them particularly susceptible to infection [2]. *Staphylococcus aureus* is one of the first pathogens detectable in the airways of young CF patients and the bacterium with the highest prevalence at an early age. Already among children younger than two years of age its prevalence exceeds 50%, reaching its peak value in the early teenage years at a prevalence of almost 80% [3,4] (Figure 1). In many patients, long term colonization with individual *S. aureus* clones is observed [5,6,7,8].

In 2018, the prevalence of Methicillin-resistant *S. aureus* (MRSA) and Methicillin-sensitive *S. aureus* (MSSA) in individuals with CF in the United States was 25% and 55.3%, respectively [3,9]. In Europe, the prevalence of MRSA is below 10% in most countries, while the prevalence of MSSA is also high, except in the UK, where continuous anti-staphylococcal treatment is part of the regular therapy regimen from the beginning of diagnosis [10]. Patients infected by MRSA are characterized by worse lung function, higher mortality, and increased hospitalization, and therefore MRSA pose a particular threat to CF patients, while there is so far no consensus on optimal treatment strategies for either MRSA or for MSSA [11,12,13,14].

Pathogens such as *S. aureus* are exposed to many hostile environmental conditions in the airways of CF patients [15]. Ecological influences such as oxygen level and nutrients, host immune defence, antibiotics and competition with other pathogens, such as *Pseudomonas aeruginosa*, cause a great selective pressure. To be able to colonize the airways and to survive in such harsh surroundings, *S. aureus* must adapt to and evolve in such unfavourable environments. How do these stressors cause adaptations, what are the evolutionary traits and what are the consequences? To address these questions, phenotypic variations need to be considered in relation to genotypic changes and the clinical impact of the adaptation processes needs to be elucidated. Once the underlying mechanisms are deciphered, targeted preventive measures can be taken and treatment options can be chosen accordingly.

This review will provide an overview about recent studies and their findings using genotypic methods such as single or multilocus sequencing (*spa*-typing or MLST) and whole genome sequencing (WGS) to identify factors contributing to the adaptation of *S. aureus* and its evolution in the airways of individuals with CF.

### 1.1. Adaptation to CF Airways

Persistent *S. aureus* clones tend to form a genetically (and especially phenotypically) heterogeneous population [15]. The genetic diversity is promoted by mutations and horizontal gene transfer (HGT). Events of *S. aureus* genome alteration have been shown to occur at significantly higher rates in isolates of CF patients than in isolates of healthy individuals [16]. The phenotypical heterogeneity of *S. aureus* populations in CF patients is intensified by differences in gene expression, which is mediated by regulatory mechanisms responding to specific milieu conditions (Figure 2) [15].

This view is supported by the finding stating that identical *S. aureus* clones are translated into differing phenotypes in distinct hosts: this suggests that gene expression is indeed influenced by the surrounding conditions, and, consequently from the latter, that individual host-factors relevantly shape the *S. aureus* adaptation process [18,19,20]. By analogy with higher mutation rates, more different *S. aureus* phenotypes are found in the airways of CF patients than in healthy subjects [21,22,23]. Genotypic as well as phenotypic changes occur time-dependently and show a longitudinal accumulation over the course of the disease [24]. The overall trend of the adaptation process seems to be directed towards a persistent and colonizing rather than infecting lifestyle and towards the establishment of a heterogeneous population, which fit into different niches in CF airways and are comparable with the “morphotypes” of *P. aeruginosa*, which describe different phenotypes [25,26,27]. This is also reflected by the fact that, even in case of an exacerbation, the presence of *S. aureus* is restricted to the airways and systemic disease is rarely observed [28,29]. The preponderance of colonizing clones results from the suppression of virulence genes for the sake of an intensified expression of adhesins and biofilm formation, granting protection against antibiotic therapy and immune-response [15,30]. This is, at least partially, due to the inactivity of the *agr* quorum-sensing system [31]. A special *S. aureus* phenotype highly trimmed towards persistence and regularly observed in CF patients are the so-called small colony variants (SCVs), which show slow growth rates, are able to persist intracellularly, feature increased resistance against antibiotics and are associated with more severe lung disease [18,32,33,34], Figure 3.

### 1.2. Clinical Impact

In young patients with CF, the presence of *S. aureus* is associated with increased inflammatory activity in the airways and a worse nutritional status [21,35,36,37]. However, there is still controversy concerning the different therapeutic approaches targeted at *S. aureus*, namely: prophylaxis, eradication or only treatment of exacerbations. This situation is reflected in the absence of internationally accepted therapeutic guidelines addressing *S. aureus* treatment in CF patients and inconsistent guidelines at national levels [38]. Prophylaxis is recommended in the UK and Australia, but not clinical practice in the US and continental Europe [39]. The main reason for ambiguity and differing approaches in therapy, which arise over the importance of antibiotic *S. aureus* prophylaxis in young children, is the shortage in valid studies and reliable data in this respect [40]. Only a few studies on this topic have been published and several of them are of poor quality [41]. Clearly, the open questions about risks and benefits of prophylaxis should be addressed in randomized controlled trials. In recent years, the need to gain knowledge in this sphere has become even more pressing: The increase in neonatal CF screening has considerably raised the number of very young patients with minimal lung disease, a condition for which effective and evidence-based treatment options are necessary [3,4]. For the status quo, Smyth et al. propose an antibiotic prophylaxis until the age of 3 years. During this period, prophylaxis seems to reduce *S. aureus* colonization rates without increasing the prevalence of *P. aeruginosa* [40]. The replacement of *S. aureus* by *P. aeruginosa* is a common process, regularly witnessed in the course of the disease (Figure 1). Nevertheless, in adult CF patients *S. aureus* prevalence still remains at about 40% and has been declining in recent years due to early antibiotic eradication regimens [3,4]. The role of *S. aureus* during the later stages of the disease is not well understood by now. Commonly, a distinction is made between *S. aureus* colonization and infection. The former mostly does not cause immediate clinical consequences and usually there is no need for therapeutic intervention, whereas the latter can lead to pulmonary exacerbations requiring treatment [42]. Until now, no bacterial characteristics have been identified in *S. aureus* that are clearly associated with infection and worse lung function [42]. However, if known, these characteristics could be of great clinical importance; they might be used to differentiate between colonizing and infecting *S. aureus* clones and thus might facilitate therapeutic decision-making.

### 1.3. S. aureus—Fundamentals of Pathogenicity

The population structure of *S. aureus* is clonal. Of all *S. aureus* strains which have been isolated from humans, 10 dominant strains (referred to as “clonal complexes” (CC)), which can be subdivided into numerous substrains, colonize, and infect humans [43,44]. Confrontation with this pathogen results in a spectrum of conditions ranging from subclinical colonization to fulminant or even lethal infections. The clinical fate is not determined by the pathogen per se, as each of the 10 clonal complexes is able to act as a harmless colonialist on the one hand and as a life-threatening pathogen on the other hand [44,45]. However, certain clones tend to be more aggressive than others [46], which raises questions about differences in the otherwise shared genetic bases accounting for *S. aureus’* differentially pronounced pathogenic potential.

The *S. aureus* genome has a size of about 26.000 kb: 75% of *S. aureus* genes are found in more than 95% of the strains and represent the bacteria’s core genome (Figure 4).

These genes are located on the chromosome and mainly encode proteins with housekeeping function, but also some virulence proteins (*spa, hlg, clf, fnb*) [47,48]. About 25% of *S. aureus* genes can be assigned to the variable genome, which is further divided into core variable genes and mobile genetic elements (MGEs).

### 1.4. S. aureus Virulence Genome

*S. aureus* exhibits an extensive exoproteome, featuring a broad repertoire of virulence genes, mainly located in the variable genome. This virulence factor pool can be roughly divided into adhesins on the one hand and toxins on the other hand (Figure 5). The expression of these proteins is controlled by a complex framework of gene regulatory mechanisms, which in turn reacts to the surrounding conditions (Figure 2) [49,50]. Cell-wall-associated adhesins are predominantly expressed during the exponential growth phase, whereas toxins are often secreted in the post-logarithmic phase [51]. This sequence of expression is considered to reflect the process of colonization, invasion and subsequent persistence. During colonization and invasion, recognition of and adhesion to surface molecules is vital, whereas in the phase of consecutive persistence a continuous nutrition supply, provided by secreted enzymes and toxins, is necessary [51]. However, this time-dependent sequence of virulence gene expression must merely be considered as a general tendency and varies between different *S. aureus* clones, diseases and hosts. Furthermore, it is still unclear if this sequence occurs during *S. aureus* colonization and infection of CF airways.

### 1.5. Role of Agr in Virulence Gene Regulation Via Quorum Sensing

Quorum sensing describes a process of cell-to-cell communication in bacteria, allowing a transfer of information concerning environmental conditions, such as bacterial density [52]. The underlying principle is the bacterial production of autoinducer peptides (AIPs). If the bacterial population reaches a certain density level, the concentration of AIPs rises and causes receptor-mediated changes in gene expression [53]. The expression of adhesins and toxins, as well as the formation of biofilms are regulated in this way by one of the most important and best understood *S. aureus* regulator-systems, the accessory-gene-regulator system (*agr*) [51]. The *agr* pathway results in a downregulation of cell wall anchored virulence factors, mainly adhesins and an induction of secreted virulence factors, mainly toxins [53]. The *agr* systems contain a variable region in AgrC, which allows the differentiation of four different *agr* types, which correlate with certain, toxin-mediated diseases; for instance, *agr*-type 3 associates with toxic shock syndrome (TSS) and *agr*-type 4 with staphylococcal scalded skin syndrome (SSSS) [54]. However, it remains unclear whether differing activities of the *agr*-types are the main causative factor determining this association or whether different *agr*-types are merely an indicator of a certain clonal background, that for its part accounts for the occurrence of the respective syndromes [45,54]. Furthermore, some *S. aureus* clones bear an inactive *agr*-system, a state referred to as *agr*-dysfunction. Corresponding to the toxin-inducing and adhesin reducing effect of *agr*, *agr*-dysfunction causes a more adherent and less toxic phenotype [55]. The downregulation of *agr* has been associated with improved persistence of *S. aureus* and therefore seems to facilitate chronic, localized infections [31,56]. In accordance with those correlations, it has been found that the *agr*-system is often inactive in CF airways, which provides a perfect example of the above-mentioned chronic, localized infection [31,57].

### 1.6. Role of Horizontal Gene Transfer in the Spread of Virulence Genes

Due to their haploid set of chromosomes, bacteria show a more direct genotype–phenotype correlation than diploid eukaryotes. This characteristic, coupled with the high pace of asexual (vertical) bacterial reproduction and mechanisms of horizontal gene transfer, allows a rapid spread of successful alleles within a bacterial population [58]. Peacock et al. observed high rates of HGT; interestingly, chromosomally localized genes also showed high transmissibility [59]. Restriction mechanisms impede DNA transfer between alien *S. aureus* strains and pose an important barrier to the emergence of highly virulent *S. aureus* clones [60]. Fortunately, due to HGT restriction no *S. aureus* strains featuring all major virulence, resistance and adaptation genes have been observed so far [61]. Whether a successfully transferred gene eventually persists in a bacterial population depends on the relation of advantages to fitness costs the gene offers in the current surroundings [62]. Considering that bacteria have a vast coding density of approximately 95%, it can be assumed that the majority of HGT events will interfere with pre-existing genes and thus will likely have adverse effects on the recipient bacterium [62]. For this reason, the process of HGT must be regulated by means of restriction mechanisms. In addition, the high likelihood of interference gives a plausible explanation for the preponderance of HGT events between related strains: in this case integration into functional networks is more probable, resulting in a relatively low risk of host-toxicity caused by the transferred gene [62]. Hence, the genetic background into which a gene is integrated decisively determines its value to the recipient [58].

Bacteriophage-mediated transduction is the most important mechanism of HGT in *S. aureus* [47,61]. β-hemolysin converting bacteriophages encoding immune evasion proteins (staphylococcal complement inhibitor (SCIN), chemotaxis inhibitory protein of *S. aureus* (CHIPS), staphylokinase (SAK)) are widespread among human *S. aureus* clones and appear to be of pathogenetic importance [63,64]. Since bacteriophages carrying immune evasion genes were found to be more prevalent in *S. aureus* isolates collected from the human host compared to isolates from animals, a role in human colonization can be assumed [65,66]. When integrating into the bacterial genome, bacteriophages interrupt the *hlb* gene and in turn provide the genes of the immune evasion cluster [64]. As a result, virulence traits can change drastically, and the bacterial population divides into various subgroups featuring different virulence gene profiles [66]. The intensity of phage-mediated transduction within a bacterial population has shown to be highly milieu-dependent. In case of environmental stress, such as antibiotic therapy and the host immune-response, it sharply increases in frequency [16,67,68]. Environmental stress does not only induce bacteriophage propagation within the population, but also reinforces the expression of bacteriophage-linked (virulence) genes [67]. Consequently, under certain conditions, the intensity of transduction can significantly influence bacterial virulence [69]. Indeed, this has been found to be the case in several toxin-mediated diseases, such as the haemolytic–uremic syndrome (HUS), toxic shock syndrome (TSS) and staphylococcal scalded skin syndrome (SSSS), in which the causing toxin-genes are phage-associated [69]. The course of these diseases can be aggravated iatrogenically by the administration of antibiotics inducing transduction, while not interfering with toxin gene expression.

In accordance with these findings, Goerke et al. demonstrated that *S. aureus* isolates obtained from CF lungs exhibit higher rates of genome plasticity than carrier isolates populating the nostrils of healthy subjects [22,57]. High HGT rates in turn are associated with increased virulence; however, it is unclear whether HGT is the cause or the consequence of virulence [62]. Without doubt, especially under selection pressure, there are HGT events which are highly beneficial for the recipient, e.g., in case of a transfer of resistance genes [62]. However, Knöppel et al. were able to show that the majority of successfully transferred genes show significant advantages neither on the level of pathogenicity and persistence, nor disadvantageous impacts on bacterial fitness, and are mostly not incorporated into the recipient’s genome on the long term [62]. This observation is referred to as “neutral diversity” and implies that a considerable number, if not the vast majority, of HGT events has no immediate impact on the recipient’s fitness. It is presumed that HGT caused “neutral diversity” reflects the possibility that some MGEs represent “selfish DNA”, as pointed out by the evolutionary biologist Dawkins, who hypothesized that the spread of MGEs lie in their own interest, but not necessarily in the interest of the recipient [70]. Nevertheless, even if “neutral diversity” lacks benefits to the single bacterium, according to the “insurance hypothesis” there might be one for the bacterial population; a heterogeneous population is more likely to successfully cope with environmental stress than a homogenous one [22,71].

### 1.7. Impact of Varying Genotypes and Single Genes on S. aureus Pathogenicity

The fact that almost all human-specific *S. aureus* isolates can cause life-threatening systemic diseases raises questions concerning the impact of the considerable amount of genetic variation and the high intensity of HGT among *S. aureus* clones on their pathogenic potential [44,46]. The current state of the research in this regard remains ambiguous. The evaluation of the influence of single virulence genes on bacterial pathogenicity is particularly challenging since single genes are embedded into a genetic background with a framework of often not fully understood expression regulation mechanisms that decisively affect gene expression (Figure 2) [57,71]. To introduce a universally accepted definition of virulence genes, Falkow et al. formulated the so called “Molecular Koch’s postulates” as follows [72]:The considered virulence gene should be associated with pathogenic clones or strains;The inactivation of the gene should measurably decrease virulence, andThe reactivation of the gene should restore pathogenicity.

Nevertheless, until now the greatest achievement in examining the impact of single virulence genes on pathogenicity was the identification of several toxin genes as a necessary requirement for certain toxin-mediated diseases. The presence of those toxin genes, however, is neither a sufficient condition for the occurrence of these diseases, nor are these genes associated with an increased overall pathogenicity [73]. On the contrary, invasive isolates often show comparatively low levels of toxin production [73]. As a matter of fact, focusing on single virulence genes as markers for overall pathogenicity can be misleading. The research addressing the role of Panton–Valentine leukocidin (*pvl*) serves as an excellent example: the pore-forming toxin has been considered a major pathogenic driver of life-threatening invasive disease, such as necrotizing pneumonia [74]. Recent research, however, revealed that there is no clear association between infections with *pvl*-positive strains and poor clinical outcomes [75]. This illustrates why it is not valid to use single virulence genes when screening for strains with high pathogenic potential, or to take eradication or treatment decisions based on the presence of these genes. These recent findings have led to the assumption that the critical parameter defining *S. aureus* pathogenicity cannot be the single virulence gene, but rather the clone or strain itself that is characterized by a unique set of virulence genes [59]. As mentioned before, according to the concept of “neutral diversity” many virulence gene encoding MGEs can be considered “selfish DNA”, for whose spread the main preconditions are a functional transfer-vehicle, such as bacteriophages, and the absence of disadvantageous fitness effects on the host. Being of advantage to the host decisively fosters a rapid propagation of a virulence gene in a population, but particularly concerning pathogenicity, it has to be considered to what extent virulence benefits a foremost commensal bacterium such as *S. aureus* in the long run. High pathogenicity, associated with fulminant disease, rapidly leads to an evolutionary dead end due and to host death [73]. In this context it must also be considered the possibility that human-pathogenic *S. aureus* strains might have a common foundation of pathogenicity located on the core genome that is only differently shaped by varying sets of core variable genes and MGEs. If this scenario holds true, intrahost conditions rather than special sets of virulence genes represent the crucial factor deciding whether colonization or infection takes place [44].

## 2. Genotyping of *S. aureus* in Cystic Fibrosis

Increasing numbers of studies are looking for answers at the genomic level in order to associate molecular alterations with changes in the clinical outcome. The characterization of *S. aureus* isolates using whole genome sequencing (WGS) enables the identification of predominant strains, allows conclusions to be drawn on occurrence, evolution, adaptation and spread and helps to control outbreaks [76,77].

First generation sequencing—or Sanger sequencing—has been used for more than 30 years to perform genetic studies in research and diagnostics and to uncover genetic variants [78]. However, the study of larger regions of the genome was not made possible until 2004 with next generation sequencing (NGS), which meant that many genes, the whole exome (WES), or the entire genome (WGS) could be sequenced [77,79]. Nonetheless, methods from both generations provide insights into the genetic basis of *S. aureus* infections and can help researchers to answer questions about population structure, evolution, pathogenicity, and virulence considering the unique circumstances in CF airways. So far, the most commonly used typing methods in studies investigating the clonal nature of *S. aureus* in cystic fibrosis are *spa*-typing [80,81,82], MLST [83,84] and, in recent times, WGS [76].

Protein A (SpA) is a ubiquitous *S. aureus* surface protein that belongs to the group of MSCRAMMs (microbial surface components recognizing adhesive matrix molecules), which mediate interaction with the extracellular matrix) which is anchored to the cell wall. Differences in the variable repeat region X are used for *spa*-sequence typing [85]. The composition of the polymorphic region (variable number of tandem repeats, VNTR) in the 3′ coding region of *spa* is analysed and the particular *spa*-type is deduced from the sequence and order of the specific repeats [86]. Individual *spa*-types can in turn be grouped into clonal complexes based on their similarity by BURP analysis (based upon repeat pattern) [82]. This can be used to trace the evolutionary lineage and degree of relatedness of strains to each other, and for identifying a common ancestor.

MLST analyses sequence fragments of seven housekeeping genes, where each strain is characterized by a seven allele profile and assigned to different sequence types (ST) accordingly [83]. The similarities between different sequence types are grouped into clonal complexes [87].

Whole genome sequencing allows the determination of differences on a single nucleotide basis and therefore enables an even higher resolution typing of strains, prediction of antimicrobial resistance and discovery of virulence factors [76,77,79]. Single nucleotide polymorphism (SNP) analysis examines the occurrence of single nucleotide changes, whereas core genome (cg) MLST looks at allelic changes in specific conserved genes (n = 1861) throughout the genome [88,89]. Whole genome (wg) MLST includes additional selected accessory loci in the analysis [90].

Genotyping of *S. aureus* has been used both in studies tracing a particular change observed in a single CF patient over time and in large groups of participants for gathering more general information on the molecular background in individuals with CF. Multicenter, longitudinal studies offer the opportunity to collect comprehensive data and associate disease progression to genotypic and phenotypic changes in *S. aureus* isolates and therefore combine both aspects. Table 1 gives an overview of recent studies using different sequencing approaches to determine the molecular basis of *S. aureus* population dynamics, transmission, acquisition of resistance and adaptation during the chronic airway infection of CF patients.

### 2.1. Adaptation of S. aureus towards Persistence in Cystic Fibrosis Using WGS

McAdam et al. were the first to investigate persistent *S. aureus* strains from CF patients by WGS [91], Table 1. They compared three isolates from one patient cultured over a period of 26 months. One of their results was that the genomes differed in phage content. The loss of the β-converting phage, which carries immune evasion proteins such as CHIPS [64], leads to the activity of ß-toxin visible as a double haemolysis zone on Columbia blood agar as depicted in Figure 3, which acts as a sphingomyelinase and is toxic to many human cells such as monocytes, neutrophils, keratinocytes and lymphocytes [92]. The authors also showed that genetic polymorphisms occurred in genes which have an impact on antibiotic resistance and global regulation of virulence. Most changes were isolate-specific suggesting the existence of a heterogeneous infection population, which evolved from a single infecting strain of *S. aureus*. Interestingly, there was a high frequency of mutations in the alternate transcription factor SigB. To investigate, if SigB is also effected in other persisting *S. aureus* strains, the authors compared DNA sequences of SigB-associated loci, such as *rsbU*, and others from 16 long-persisting *S. aureus* strains from four other CF patients [7]. However, they could not confirm elevated changes in the other strains. From these results, the authors concluded that the elevated mutation rate in SigB associated loci is related to the genetic background of the strain or to patient specific selective pressures.

Long et al. conducted a longitudinal multicentre study and examined 1382 collected respiratory *S. aureus* isolates of 246 children with CF treated at 5 US CF centres over a mean period of 2.2 years using WGS with 5.6 *S. aureus* isolates per patient (range 2–22), [93]. With their analysis, the authors intended to investigate the population dynamics of *S. aureus* and to figure out if there are special genes prone to adaptation. WGS revealed that in their cohort of patients, 45.5% of patients carried multiple coexisting lineages with different resistance profiles. It would have been interesting to know if any of the coexisting lineages persisted during the study period or if they were lost, but this was not reported by the authors. However, more than 50% of isolates belonged to 3 MLST-types (ST5, 8 and 30), with ST5 representing a well-known healthcare-associated MRSA lineage and ST8 a community-associated MRSA lineage. Only few strains were occasionally shared between individuals belonging to the same household or due to clinical exposure. Some MLST-types were distributed throughout all centres and included MRSA as well as MSSA, which resembles the *S. aureus* carriage profile in the study of Westphal et al. described as “prevalent” [94]. In addition to the conserved core genes, the authors analysed the variable accessory gene content. Interestingly, 655 core genes were present in more than 99% of isolates and 1142 in more than 95% of isolates, respectively. Many unique genes (n = 21,358) were identified in the pangenome for the entire isolate collection. From the number of new genes which were identified per genome, the authors concluded that *S. aureus* has an open pangenome with an ongoing flow of new genetic elements coming into the *S. aureus* populations. The in vivo mutation rate for the investigated isolates (including SCVs) was comparable to that of general *S. aureus* populations, contrary to their assumptions and the observations by Besier et al. [95]. An increase in mutation rates was only observed for isolates with *mutS* or *mutL* non-synonymous mutations. Strains with such mutations are often cultured during persistent infection in CF and can lead to a hypermutator phenotype, as shown for *P. aeruginosa* in particular, but also for *S. aureus*, which is associated with trimethoprim-sulfamethoxazole and macrolide resistance [95,96,97,98]. Furthermore, it could be shown that the reason for the genetic diversity in patients was mainly due to the simultaneous infection with multiple *S. aureus* lineages, which has also been observed by Ankrum et al. and Westphal et al., described here as the “dominant” carriage profile [94,99]. An excess in non-synonymous mutations was detected in nine genes indicating a positive selection during adaptation to the CF airway environment. Five of the genes (*thyA, rpoB, rspH, fmtB*, and *walK*) were associated with antibiotic resistance. Consistent with *agr*-dysfunction, isolates with mutations in the virulence regulators *agrA* and *rsbU*, displayed changes in virulence factor expression such as haemolysis, pigmentation and protease activity. The authors argued that such a positive selection with the loss of these regulatory functions in these important regulators favours a phenotypic change from acute to chronic infection.

Lange et al. investigated isolates obtained during a longitudinal, multicentre study, which included 183 CF patients from 16 centres in Germany and one centre in Austria during a period of 21 months. They collected 3893 *S. aureus* isolates, which they characterised by *spa*-typing, with a mean number of 21 isolates per patient (range 1–83) [94]. Furthermore, PCR analysis of 25 important virulence genes (*cap 5/8, chp, cna, clf A/B, sdrC/D/E, fnbA/B, sasG/H, eap, emp, sea–sej, eta/etb, tst-1, pvl, hlg* and the four *agr*-specificity groups I to IV) was performed [100]—see Table 1. No association could be established between specific virulence genes and the clinical status, but *agr*-type IV was associated with poorer disease outcome. The authors hypothesized a connection between the phylogenetic background of a clone and the *agr*-system, in which the latter might play an indicator role, as has also been shown by Jarraud et al. [54]. This was supported by the observation that the prevalence count of specific virulence genes could be unambiguously assigned to an *agr*-type, as could each predominant *spa*-type. Additionally, this analysis showed that there was a time-dependent loss of virulence genes, which the authors concluded was most likely due to *S. aureus* adaptation towards persistence and colonization, as virulence factors are potential targets for immune response. In this study, environmental stress, which was defined by antibiotic therapy and/or immune response as indicated by an increase in IL-6, most likely favoured HGT, as the rate of virulence plasticity increased under these conditions, indicating that the *S. aureus* virulence genome changes due to an impact of iatrogenic and host-specific factors.

Azarian et al. [101] analysed 85 MRSA strains from four CF patients with long-term persistence. An average of 21 isolates per patient (range 19–23) with a mean persistence of 3.8 years (range 2.5–4.6 years) were subjected to WGS to study intrahost genomic diversity, population structure, evolution history, evidence of switched intergenic regions (IGR) and signatures of adaptation. The authors show that a dominant lineage of *S. aureus* persisted together with transient additional lineages, which is in line with the studies by Long et al. [93] and Westphal et al. [94]. There was considerable intrahost genomic diversity and also variations in antibiotic resistance, even in isolates collected on the same day. Within-host nucleotide diversity varied among patients, but were in the range of other reports between 10^−7^ to 10^−5^ SNPs per site per year. By diversifying the selection of genes across clonal intrahost populations, 11 genes were identified in three of the four studied individuals. Some of the altered genes (*odhA, mprF, clpX*) have already been associated with persistence (e.g., biofilm formation) and attenuated virulence. Only one gene, *spa*, was found to have evidence of genewide diversification selection in all patients. The authors show that intrahost adaptation is a continuous process despite purifying selective pressure. However, a limitation of the study is the fact that three of the studied patients were infected by relatively closely related ST5 populations, which could have led to an underestimation of evolutionary events.

Langhanki et al. investigated 21 *S. aureus* isolates from the respiratory tract of a single CF patient over the course of 13 years [102] by WGS, which separated 7 different lineages, which is in line by other studies [93,99,101]. Importantly, the isolates collected during long-term persistence allowed them to study two successfully persisting lineages, which belonged to *spa*-types t012 and t021. WGS revealed that these isolates were closely related, suggesting that the t021-lineage evolved from the t012-lineage. This hypothesis was corroborated by a nearly identical set of mobile genetic elements detected in both lineages. During the transformation from t012 to t021, an increase in genomic changes, including HGT from other *S. aureus* lineages, occurred. Such events are in line with the observations of Long et al., both in terms of the presence of several *S. aureus* lineages and also due to the finding that *S. aureus* has an open pangenome, with new genes coming into existing *S. aureus* populations [93].

In their study, Herzog et al. investigated the interaction of *S. aureus* with neutrophils regarding their potential to escape from neutrophil extracellular traps (NETs) [103], which kill bacteria by antimicrobial peptides residing on expelled DNA [104]. However, *S. aureus* secretes nuclease, which can degrade NETs, thereby facilitating escape from NET-killing [105]. Herzog et al. hypothesized that during the persistence of *S. aureus*, the bacteria would adapt to NET-mediated killing. To study this, the authors analysed 111 *S. aureus* isolates from one CF patient over a period of 14 years regarding nuclease activity using a nuclease FRET assay [106]. After 11 years of persistence, isolates with increased nuclease activity were selected, which survived significantly better during NETosis compared to isolates with unchanged nuclease activity [103]. WGS of seven isolates of this patient was used to confirm the relatedness of strains and to analyse the underlying mechanism for increased nuclease activity. While in laboratory strains, nuclease has been shown to be regulated by the two-component system SaeRS [107,108], this regulator was not affected in the clinical isolates. However, isolates with high nuclease expression carried two non-synonymous SNPs in *agrA*. In line with this, an inverse association of *agr* transcription, as assessed by the transcription of its effector molecule, RNAIII and high nuclease expression was established, indicating *agr*-dependent regulation of nuclease.

Tan et al. sequenced 3 pairs of early/late *S. aureus* isolates from CF-patients with long-term persistence (mean 6 years, range 3 and 9 years) by WGS [6]. In addition, the authors tested biofilm formation, intracellular persistence, proteomic and metabolic profiles. For example, increased biofilm formation and intracellular persistence of late isolates was most likely due to mutations in *fakA*, fatty acid kinase A, which has been shown to be important for biofilm formation in MRSA [109]. Additionally, mutations in *agrC* are responsible for the downregulation of toxins and upregulation of adhesins leading to increased intracellular persistence, while mutations in *thyA* conferred resistance to trimethoprim-sulfamethoxazole and the thymidine-dependent SCV phenotype [110,111]. Some such mutations have also been shown in the study of Long et al., as genes that show a convergent evolution in CF-*S. aureus* isolates, which is also supported by the data of Tan et al. [93].

In a recent study, Haim et al. investigated the intra- and interhost diversity of 20 MRSA isolates collected from four CF patients within two short periods [112]. Genetic alterations were found both in the core and the accessory genomes, with independent mutations in *thyA, htrA, rspJ* and *gyrA*. The authors concluded that mutations in these genes can be associated with benefits for survival, and adaptation, namely antimicrobial resistance, virulence, iron scavenging, and oxidative stress.

### 2.2. S. aureus Population Dynamics in CF

Ankrum et al. used WGS of 301 *S. aureus* isolates recovered from 115 CF patients from a single centre during a period of 22 months to explore the population dynamics and relatedness of strains, to uncover transmission events and to determine a genomic molecular clock [99]. The number of differences in single nucleotide polymorphisms between pairs of isolates was used to rank the degree of relatedness and to reveal transmission events between siblings. In addition, the prediction tool PPFS2 was largely successfully used to predict MRSA phenotypes, which was mainly attributed to the gain of *mecA*. It was found that the phylogenetic order did not match the chronological order in which the isolates were collected, suggesting copopulation of different strains at the same point in time.

Westphal et al. investigated the dynamics of *S. aureus* in individuals with CF persistently colonized with *S. aureus* by *spa*-typing 3893 isolates from 183 patients from 17 centres throughout Germany and one centre from Austria during a 21-month period [94]. Four *S. aureus* carriage profiles could be observed: 1. Patients were considered to have prevalent *spa*-types (n = 68) if isolates with the respective *spa*-types were isolated from at least 10 other patients from the same or other centres and isolates with the prevalent *spa*-type were in at least 50% of respiratory specimens of the patient; 2. Patients were considered to have one *spa*-type (n = 64), if the isolates mostly belonged to only one *spa*-type. To be assigned to this group, patients were allowed to carry isolates of only one other respiratory specimen belonging to another *spa*-type. 3. Patients were considered to have a dominant *spa*-type (n = 65) if one *spa*-type was traceable throughout all visits but isolates belonging to other *spa*-types also occurred. 4. Patients we considered to have related *spa*-types (n = 33) if the repeat region of the *spa*-type indicated that mutations including insertions, deletions or point-mutations occurred. The carriage profiles of patients were associated with age, antibiotic treatment and lung function. While antibiotic treatment was associated with patients belonging to the “one *spa*-type” carriage profile, patients with “dominant” or “related” *spa*-types were significantly older [94]. The clonal relatedness of the isolates of individual patients with related *spa*-types was confirmed by WGS. Twenty-four isolates from nine patients were sequenced by WGS and compared by using their cgMLST to corroborate the close relationship of isolates with related *spa*-types for every patient. Deletions in the VNTR region are ultimately beneficial for the persistence of *S. aureus* since SpA induces the inflammatory response. It has been shown by Garofalo et al. that a shorter VNTR region is associated with a lower degree of immune response [113]. Additionally, the increasing predominance of only “one *spa*-type” in CF patients with higher antibiotic treatment burden indicates the selection of a specific clone and the prevention of the acquisition of other, new clones in response to anti-staphylococcal treatment.

Ormerod et al. took another approach and used WGS to compare the genomics of non-pseudomonal bacterial species present in cystic fibrosis patients, including six *S. aureus* isolates from three CF patients from broncho-alveolar lavage [114]. They sequenced four sequential isolates from patient A and one isolate from patients B and C, respectively. All isolates belonged to different MLSTs, except isolates three to five from patient A, which were closely related suggesting persistence for one year. All isolates carried the *blaZ/blaI/blaRI* operon conferring penicillin-resistance, but were susceptible to a wide range of antibiotics. Only strains three to five from patient A carried the CHIPS gene *chp*, while strain one from patient A contained more enterotoxin genes and had a deletion in the capsule gene, *capH*.

### 2.3. MRSA in Cystic Fibrosis—Antibiotic Resistance, Persistence and Patho-Adaptation

Methicillin resistance is attributed to the acquisition of the *mecA* resistance cassette, which encodes an altered penicillin-binding protein (PBP2a) and results in resistance to all β-lactam antibiotics available [115]. The staphylococcal cassette chromosome *mec* (SCC*mec*) is a mobile genetic element and horizontal gene transfer between different species is suspected due to its conserved nature [116]. Variability within the *mec* gene complex suggests frequent genetic exchange and, to date, at least 12 types of SCC elements have been described [115,116]. Hospital-acquired (HA-) MRSA tend to carry additional resistance genes on SCC*mec* and are often type I, whereas community-acquired (CA) MRSA, usually type IV or V, usually have only one additional resistance but are frequently Panton-Valentine leukocidin (PVL) positive [117]. Additionally, livestock-associated MRSA have recently been reported to infect CF patients [118,119]. LA-MRSA have been reported in food-producing animals and in humans exposed to such animals [65], but rarely in CF patients.

Porterfield et al. investigated 97 MRSA isolates from 74 individuals with CF from 2017 and determined that MLST types showing ST5 (n = 55), ST8 (n = 15) and ST105 (n = 14) were the most prevalent [120]. Closely related MRSA strains were only shared by family members and rarely between unrelated individuals. The analysis of the accessory genome content of all isolates allowed them to distinguish three different clusters, which differed in age and clonal distribution. Cluster A was present in all age groups, while cluster B was more prevalent in patients younger 20 years. Patients with isolates belonging to cluster A had lower lung function and higher sputum biomass compared to patients infected by cluster B isolates.

Several studies used WGS to unravel the mechanism of linezolid or tedizolid resistance:

Antonelli et al. used WGS to sequence a linezolid-resistant MRSA from a CF patient and analyse the resistome, revealing that the strain carried the *cfr* gene, which is responsible for linezolid resistance [121], and other resistance genes being consistent with the phenotypic resistance profile [122]. Obviously, the *cfr* gene must have been acquired by the MRSA strain recently since earlier MRSA isolates were linezolid susceptible. Unfortunately, earlier MRSA strains were not available for further analysis.

Rouard et al. identified the underlying mechanism of linezolid resistance in MRSA isolates of a patient infected by MRSA for 15 years [123]. The authors sequenced 13 MRSA isolates of this patient collected over 5 years by WGS. The patients was repeatedly treated with linezolid which resulted in rapid acquisition of high-level resistance. Sequencing revealed a specific mutation (G2567T) in all five copies of the 23S RNA gene (*rrl)*, which was associated with a strong increase in generation time. In addition, varying numbers of altered *rrl* copies matching the course of treatment were found and positively linked to the MIC.

Boudet et al. studied the emergence of linezolid-resistance in two CF siblings with persistent MRSA infection by analysing 17 isolates by WGS [124]. Both patients had previously been treated with more than 10 courses of linezolid therapy over 15 days. After the first emergence of a linezolid-resistant strain, in all subsequent samples linezolid resistance appeared. However, if up to five morphotypes per sample were investigated, linezolid-susceptible strains were also, observed documenting intrasample heterogeneity of susceptibility. Four linezolid-resistant strains were subjected to WGS, which revealed a unique mutation (G2576T substitution) that was not present in the susceptible isolates. SNP-based analysis revealed highly diversified strains which, according to the classification by Ankrum et al. [99], were clonally related.

Roch et al. studied the susceptibility to tedizolid, a new oxazolidinone with substantial lung penetration, of 330 *S. aureus* isolates from three CF centres [125] and detected two isolates from two different centres with values above the FDA breakpoints. Both isolates were sequenced and revealed mutations in the domain V region of the 23S rRNA gene, *rrl*, a G2576T mutation, which has been also found by Boudet et al. in a linezolid-resistant strain [124]. Additional mutations in other ribosomal protein-associated genes, *rplA* and *rpsQ*, were found, which have not been characterized in terms of functionality.

Rolain et al. sequenced one MRSA isolate of a clone named CF-Marseille, because this clone has been shown to spread among CF-patients [126]. In their retrospective analysis, 25 of 80 patients infected by MRSA showed the special phenotype of this clone, which grows on Cepacia agar with intense orange pigmentation, has a thickened cell wall in electron-microscopy and displays a specific susceptibility profile with resistance to tobramycin, ciprofloxacin, cotrimoxazole and heterointermediate resistance to vancomycin. WGS of one isolate of this clone revealed the presence of a new mobile genetic element, a new SCC*mec* type IV mosaic cassette that has become integrated into a plasmid and a new phage.

López-Collazo et al. investigated the evolution of MRSA during long-term persistence and the interaction of early and late MRSA isolates with the host [127], Table 1. The authors analysed the genetic evolution of a single clone of an individual CF patient by sequencing an early and a later isolate recovered 13 years apart, which resulted in both reduction of the bacterial chromosome and decrease in the immune response of the patient, as determined by challenge of the patient’s monocytes with the two strains. From these results, the authors concluded that the attenuation of virulence with decreased immune response establishes an equilibrium and facilitates the long-term persistence of *S. aureus*.

Long et al. elucidated the underlying molecular mechanism in a MRSA isolate with high ceftaroline resistance [128], which was isolated from the blood of one individual CF patient. Using WGS, the authors identified two SNPs in the adjacent codons of the gene encoding PBP2a. The mutations resulted in the change of two amino acids in the ceftacaroline binding pocket, leading to the development of resistance. The authors confirmed the function of this mutation by construction of a mutant, which carried this mutation. Furthermore, mutations in *thyA* were discovered that supposedly resulted in the formation of the observed SCV phenotype [110].

### 2.4. Biofilm Formation—An Adaptive Mechanism of S. aureus in Cystic Fibrosis

Biofilm formation has been described as an important virulence trait and mechanism for the adaptation of *S. aureus* in CF airways as it facilitates immune evasion and helps to protect the pathogen from antibiotic treatment [129,130]. Similarly, mucoidity and biofilm forming in *P. aeruginosa* strains have been associated with long-term persistence and worse disease outcomes [131].

Schwartbeck et al. [132], Table 1, investigated the underlying mechanism for mucoid isolates, which were isolated from some CF patients during long-term periods. The authors determined the prevalence of mucoid isolates retrospectively from the microbiological database (8 out of 313 CF-patients, 2.5%). Mucoidity was linked to strong biofilm formation in a microtiter plate assay and was associated on the genomic level with a 5 bp deletion in the intergenic region between *icaR* and *icaA.* After years of persistence of mucoid isolates, non-mucoid isolates with a 5bp deletion emerged, which carried compensatory mutations in other *ica* genes [132]. WGS of the sequential isolates of two patients with long-persisting mucoid isolates showed that the isolates within the individual patients clustered closely together, while the isolates between the patients did not.

Gabryszewski et al. analysed 12 MRSA isolates collected within a two to four-year time period from three CF patients by WGS [133]. The authors showed that the adaption process is accompanied by the acquisition of mutations in the genes that affect the staphylococcal metabolism. Mutations in *dagA* and *gpdP*, which are involved in the synthesis and hydrolysis of cyclic-di-adenosine monophosphate (c-di-AMP), have been associated with an increase in biofilm formation. Alterations in the cell wall structure were attributed to mutations in *ltaS*, which encodes a lipoteichoic acid synthase, resulting in susceptibility to Congo Red, which inhibits the activity of *ltaS*. Furthermore, mutations in genes involved in the tricarboxylic acid (TCA) cycle, most significantly *fumC*, which resulted in the use of different carbon sources have been shown to positively impact the biofilm formation.

Bernardy et al. examined in an in-depth study 64 *S. aureus* isolates from 50 CF patients for their differences at the genomic level as assessed by WGS [134] and their phenotypes in terms of haemolysis and growth on Congo Red agar, as well as the interaction with mucoid and non-mucoid *P. aeruginosa* [135]. While the toxin genes *hla* and *hlb* were present in all isolates, the haemolytic phenotype did not entirely correspond. The authors defined a mucoid phenotype solely by evaluation of colonies on Congo Red agar and inferred from this a polysaccharide production for 69.2% of the isolates. However, the authors did not find the 5-bp deletion in the intergenic region as demonstrated by our group for real mucoid isolates, which were present in much smaller percentage [132,136] and displayed another growth on Congo Red agar, as shown in Figure 3. Therefore, it is not clear if the isolates of the study by Bernardy et al. are really PIA overproducers. This should be validated for example using microtiter biofilm experiments. Furthermore, the authors characterized the isolates in terms of their interaction with non-mucoid and mucoid *P. aeruginosa* laboratory strains, which allowed them to differentiate three groups of interaction: group 1, isolates killed by non-mucoid *P. aeruginosa*; group 2, isolates killed by non-mucoid and mucoid *P. aeruginosa* and group 3, not killed at all. However, there was no genetic relationship of the different interaction groups. The authors suggested that the group 1 isolates may come from initial infection, because these isolates were mainly cultured from younger patients. The group 2 isolates were non-haemolytic, but were more resistant to antibiotics, indicating that these isolates cause more chronic infection due to the loss of expression of virulence genes. The group 3 isolates were from patients being coinfected with *P. aeruginosa* indicating that these isolates acquired defence strategies that allow them to coexist with *P. aeruginosa*.

Tomlinson et al. investigated the immunometabolic response of *S. aureus* [137]. Their data showed that *S. aureus* induced the production of the immunoregulatory metabolite itaconate in airway immune cells by stimulating mitochondrial oxidative stress. Itaconate, in turn, inhibited *S. aureus* glycolysis and growth and directed carbon flux through bacterial metabolic pathways, thereby increasing biofilm formation. These results of itaconate-induced metabolic changes were mirrored in longitudinal isolates of an individual patient being infected by *S. aureus* for 14 years. WGS of these isolates revealed several SNPs and mutations in various genes responsible for metabolism, virulence factors and regulators including *agrA*.

**Table 1 pathogens-10-01177-t001:** Recent studies using sequencing approaches.

Study Authors	Number of Patients	Length of Study [Years]	Number of *S. aureus* Isolates for WGS	Sequencing Techniques	*S. aureus* Methicillin Resistance [%]	Most Important Outcome
Antonelli et al., 2016 [122]	1	N/A	1	WGS SCC*mec*-typing	100	- the MRSA isolate was linezolid resistant and carried a *cfr* gene. The strain belonged to the ST5-MRSA-II lineage *spa*-type t306
Ankrum et al., 2017 [99]	115	1.8	301	- WGS - cgMLST -SCC*mec* typing	70	- definition of relatedness of strains by WGS: same strains: less than 71 SNPs very closely related strains: between 72 and </= 123 SNPs closely related strains: between 124 and </=156 SNPs distantly related strains: between 157 and </= 1514 SNPs unrelated strains: between 5957 and 21,644 SNPs- prediction tool PPFS2 was successful in predicting MRSA phenotype- Sequential isolates of patients did not follow a molecular clock (multiple strains at the same time that were not isolated?)
Azarian et al., 2019 [101]	4	3.8	85	- WGS - cgMLST - *spa*-typing	100	- peristence of a dominant lineage- intrahost diversity: variation in antibiotic resistance, mutations in eleven genes of three patients, *clpX,* *odHA, fmtC, mfd* among others- rates of molecular evolution between 2.21 and 8.64 nucleotide polymorphisms per year- gene-wide diversifying selection of *spa*
Bernardy et al., 2019 [134]	50	N/A	65	- WGS		- sequences are provided online - no further analysis performed
Bernardy et al., 2020 [135]	50	N/A	64	- WGS - MLST - *spa*-types - *agr*-types	61	- variety of genotypes, not a specific “CF-clone”, CC5 and CC8 most common - CC8 clones carried PVL - between patients no transmission - all isolates carried alpha- and beta-toxins and many other virulence genes - isolates group together in terms of polysaccharide production - genetic association based on the status of coinfection with *P. aeruginosa*- close relatedness of longitudinal isolates sharing similar phenotypes
Boudet et al., 2021 [124]	2	7	17	- WGS	100	- highly diversified clonally related populations - resistance to linezolid due to a G2576T substitution in a variable number of 23S rRNA gene copies - intrasample heterogeneity of linezolid susceptible/resistant strains
Gabryszewski et al., 2019 [133]	3	4	12	- WGS	100	- association of persistent MRSA infection with staphyloccocal metabolic change- altered carbon metabolism: adaption by selective use of tricarboxylic acid cycle and biofilm formation- increased transcription of specific metabolic genes, most significantly *fumC* associated with decreased induction of proinflammatory cytokines
Haim et al., 2021 [112]	4	0.7	20	- *spa*-typing (Sanger) - MLST (Sanger) - WGS	100	- intrahost diversity- genetic differences in sequential isolates in both core and accessory genomes - independent mutations in *thyA, htrA, rspJ* and *gyrA*- non-synonymous mutations in genes associated with antimicrobial resistance, virulence, iron scavenging and oxidative stress resistance
Herzog et al., 2019 [103]	1	14	7	- *spa*-typing (Sanger) - WGS/cgMLST		- inverse expression of nucleases and *agr* during long-term adaption- high nuclease activity facilitates survival with NETs- mutation in *agr* most likely responsible for increased nuclease
Lange et al., 2020 [100]	195	1.75	-	*- spa*-typing (Sanger) (3180) - multiplex PCR for adhesin genes: *clfA/B,cna, eap,* *emp,fnbA/B, sasG/H,* *sdrC/D/E* toxin genes: *eta,etb, hlg, pvl,* *sea-j,tst* immune evasion genes: *chip, cap5/8*		- association of *agr*-type 1 and 4 with clinical status- virulence gene count varies with *agr*-type- loss of virulence genes during persistence- increased virulence gene plasticity in the presence of environmental stress (antibiotic therapy/inflammation)
Langhanki et al., 2018 [102]	1	13	75	- *spa*-typing (Sanger) - MLST - WGS		- several lineages were present, but only two lineages co-existed for extended periods - genomic elements are transferred between lineages- increase in genomic changes - adaptation of successful lineages and elimination of inferior lineages
Long et al., 2014 [128]	1	0.4	6	- WGS	100	- 2 SNPs in PBP2a leading to ceftaroline resistance- mutation in *thyA* leading to the SCV phenotype
Long et al., 2021 [93]	246	2.2	1382	- WGS -wgMLST - SCC*mec*-typing (WGS)	28	- frequent co-infection by multiple, genetically distinct *S. aureus* strains often differing in antibiotic susceptibility- identification of common set of *S. aureus* genes frequently mutated during long-term adaptation (*rpoB,fmtB, agrA, rpsJ, rsbU, thyA,* *set9, walK, ebh*)- contribution to antimicrobial resistance (*rpsJ*) and bacterial persistence (*agrA, rsbU, ebh*) or both (*rpoB,thyA*) - open pangenome with incoming new genes
Lopez-Collazo et al., 2015 [127]	1	13	2	- WGS SCC*mec*-typing (Sanger) - *spa*-typing (Sanger)	100	- reduction of bacterial chromosome - loss of one of three phages in the late isolate compared to the early isolate - decrease in immune response
McAdam et al., 2011 [91]	1	2.2	3	- WGS		- variation in phage content - genetic polymorphism in genes, which impact antibiotic resistance and global regulation of virulence - genetic variations correlated with phenotypic changes in terms of hemolytic activity, antibiotic sensitivity, growth rate
Ormerod et al., 2015 [114]	3	1	6	- WGS - MLST	0	- all patients carried different strains - persistent infection in one patient - different virulence factor profiles in strains
Porterfield et al., 2021 [120]	74	N/A	97	- WGS	100	- most prevalent ST5 (HA-MRSA) and ST8 (CA-MRSA) - closely related MRSA strains only shared by family members - accessory gene content discriminates 3 clusters Cluster A: including ST5 and ST105, highly prevalent at all ages Cluster B: including ST8, younger patients Cluster C: only 6 distantly related strains - patients infected by cluster A had lower lung function and higher sputum mass compared to age matched patients infected by cluster B
Roch et al. 2019 [125]	2	N/A	2	- WGS		- underlying mutations for tedizolid resistance identified - G2576T mutation in *rrl* and new mutations in *rplA* and *rspQ* - functional experimental validation of the new mutations necessary
Rolain et al. 2009 [126]	1	N/A	1	- WGS	100	- CF-Marseille clone with a hetero-glycopeptide Intermediate resistance - discovery of a new SCC*mec* type IV mosaic cassette and a new phage
Rouard et al., 2018 [123]	1	5	7	- *spa* typing (Sanger) - MLST (Sanger) - wgMLST	100	- G2576T mutation in all 5 *rrl* copies leading to linezolid resistance - correlation of MIC and mutated copies of *rrl*
Schwartbeck et al., 2016 [132]	2	1.2	12	- WGS - cgMLST		- 2.5% of patients with mucoid *S. aureus* isolates - closely related strains within an individual patient, but not between patients - 5bp deletion in the intergenic region of the *ica* operon responsible for the mucoid phenotype
Tan et al. 2019 [6]	3	6	6	- WGS - MLST		- long-term persistence of the same clone in patients - identification of underlying genetic modifications that induce protein expression profiles and metabolic changes of late isolates compared to early isolates (biofilm, SCV, intracellular persistence) - changes were identified in e.g., *agrC, saeR,* *thyA, gyrB, fakA*
Tomlinson et al. 2021 [136]	1	14	7	- WGS		- *S. aureus* induces an itaconate-dominated immunometabolic response in immune cells - itaconate inhibited *S. aureus* glycolysis and redirected carbon flux to the formation of polysaccharide biofilm - longitudinal isolates revealed numerous SNPs in virulence genes and regulators and genes involved in metabolism
Westphal et al., 2020 [94]	8	1.75	24	- *spa*-typing (Sanger) (3893 isolates 183 patients)		- no association of *spa*-type and lung function - association of antibiotic therapy and age on carriage profile of *spa*-types during persistence- antibiotic treatment most likely prevents acquisition of new *S. aureus* clones - probability of dominant or related *spa*-types increases with age - related *spa*-types are mainly caused by deletions in the VNTR region - isolates with related *spa*-types belong to a clonal lineage as confirmed by WGS

## 3. Summary and Conclusions

In our review, we aimed to summarize the results of all the available studies that used WGS of *S. aureus* CF isolates to address a variety of research questions such as determining population dynamics, confirming the persistence of a single *S. aureus* lineage during chronic infection, elucidating the patho-adaptation during persistence, identifying the underlying mechanisms of metabolic or phenotypic changes, or new resistance mechanisms.

The results of the studies provide results which are in agreement on some points but differ on others:There is not “a” particular CF clone, but CF patients are colonized and infected by various *S. aureus* clones, which are also common in both healthy and diseased humans.CF patients are infected by coexisting lineages; however, a dominant clone largely persists.There is limited transmission of clones within the CF population.The sequencing of long-term persistent sequential isolates reveals evolution and adaptation of isolates, some of them being convergent, especially as mutations in important virulence regulators such as *agr* and *sigB*, were observed. Additionally, mutations in *thyA* were often present especially in centres where patients are treated with trimethoprim-sulfamethoxazole.During persistence, *S. aureus* changes from an acute virulent to a chronic non-virulent isolate.*S. aureus* has an open pangenome with new incoming genes from coinfecting lineages.

Since *S. aureus* is nowadays the most commonly isolated pathogen occurring the airways of CF patients in many countries, it is expected that more sequencing studies will be performed. For these upcoming studies, it will be important to optimize sequencing in terms of investigating several isolates from single visits and also from long-term persisting and sequential isolates. This will allow us to understand the diversity of *S. aureus* and their evolution during long-term infection for the development of more effective treatment, which might lead to personalized treatments depending on the observed adaptation of *S. aureus* lineages.

## Figures and Tables

**Figure 1 pathogens-10-01177-f001:**
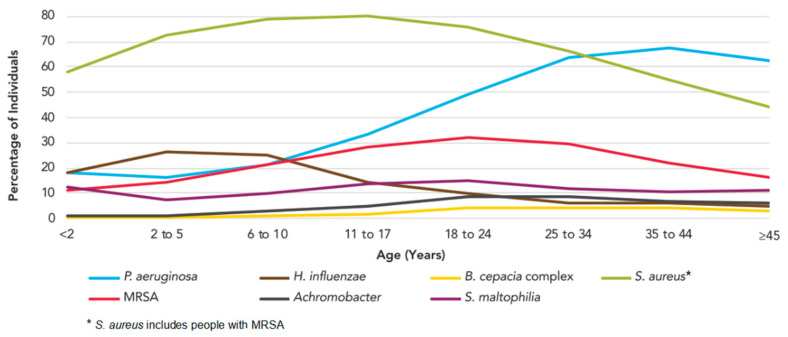
Age-dependent prevalence of different bacteria in CF patients (adapted from Cystic Fibrosis Foundation “2019 Patient Registry Annual Data Report” based on the data of 31.199 patients [3]).

**Figure 2 pathogens-10-01177-f002:**
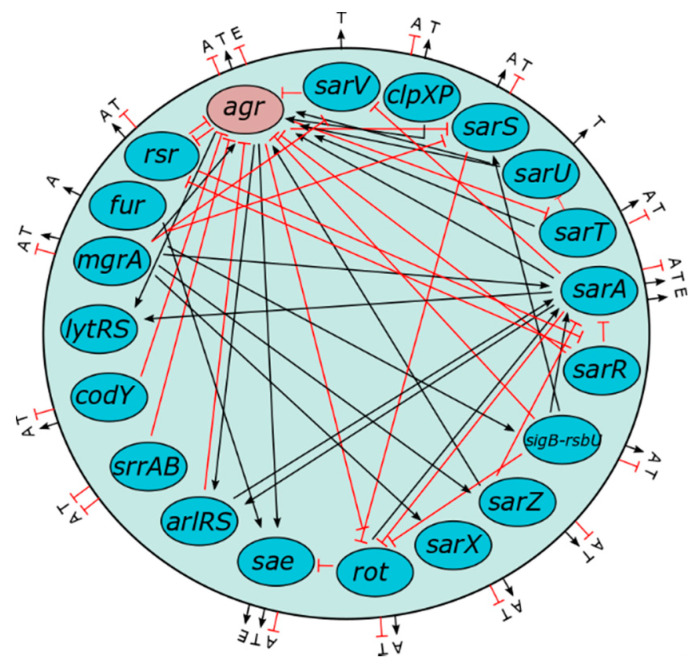
*S. aureus* gene regulatory network. S. aureus regulatory genes, their reciprocal interference and their impact on adhesiveness (A) toxicity (T) and evasiveness (E) (adapted from Priest et al. [17]). Gene expression in *S. aureus* is regulated by a highly complex and not fully understood network of genes which mutually influence their activity (black arrow: induction, red arrow: inhibition).

**Figure 3 pathogens-10-01177-f003:**
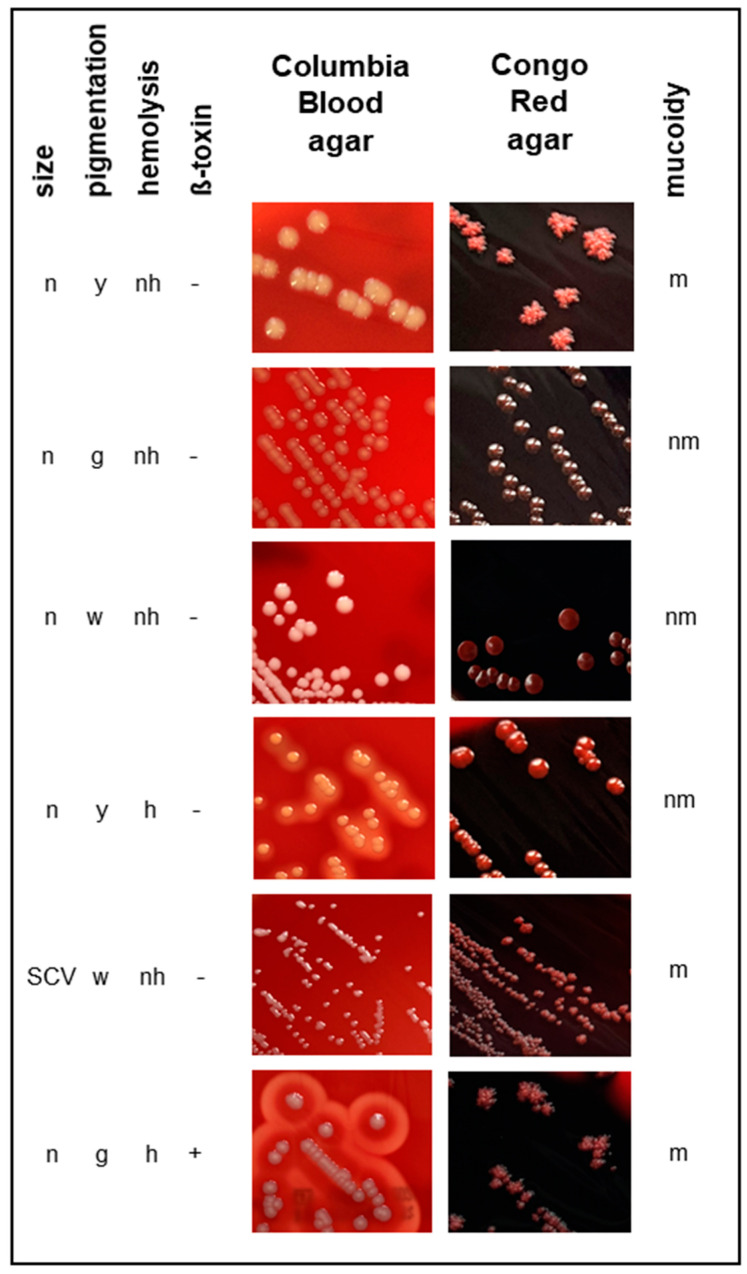
Images of different *S. aureus* phenotypes, which are often present in CF specimens. *S. aureus* strains were cultured on Columbia Blood agar and Congo Red agar (facilitates the discrimination of non-mucoid and mucoid isolates) plates. They are characterized by the following parameters: size, pigmentation, haemolysis, ß-toxin and mucoidy. Size: normal (n), small colony variant (SCV); pigmentation: yellow (y), grey (g), white (w); haemolysis: haemolytic (h), non-haemolytic (nh), ß-toxin: negative (−), positive (+); mucoidy: non-mucoid (nm), mucoid (m).

**Figure 4 pathogens-10-01177-f004:**
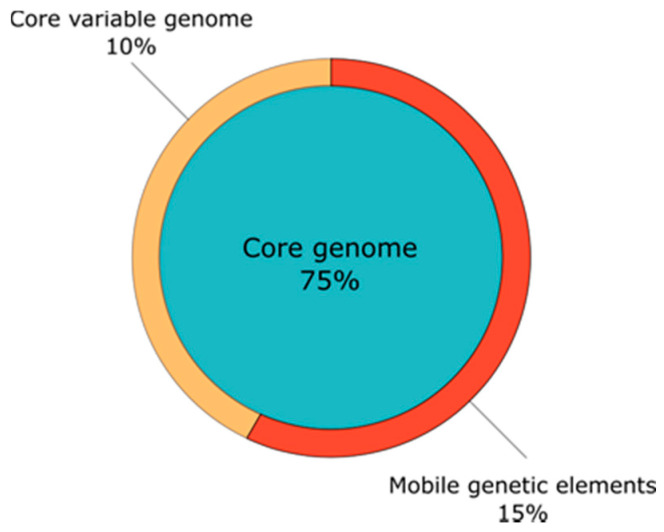
*S. aureus* genome overview: 75% of *S. aureus* genes are found in more than 95% of the strains and represent the bacteria’s “core genome”. The distribution of “core variable genes” differs between varying *S. aureus* strains. Mobile genetic elements (MGEs) such as bacteriophages, plasmids and pathogenicity islands, mainly encode for resistance proteins, superantigens and toxins. In addition to vertical transfer, they can be transmitted horizontally [44,45].

**Figure 5 pathogens-10-01177-f005:**
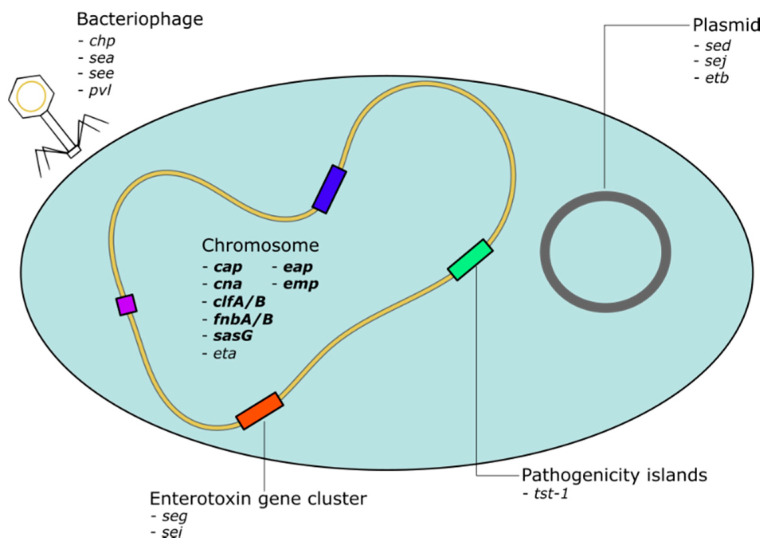
*S. aureus* virulence genes (selected) and their localization. Virulence factors of *S. aureus* are localized on different sites. While many of the adhesin genes (in bold italics) are located in the chromosome), toxins are often located on plasmids, pathogenicity islands or carried and transduced by phages.

## Data Availability

Not applicable, since all reviewed studies provided this statement, if necessary.

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
