# Peer review of "Staphylococcus aureus and Cystic Fibrosis—A Close Relationship. What Can We Learn from Sequencing Studies?"

_pathogens, 2021, doi:10.3390/pathogens10091177_

Round 1

Reviewer 1 Report

The article by Christine Rumpf et al. entitled “Staphylococcus aureus and cystic fibrosis – a close relationship. What can we learn from sequencing studies?” is a review article that aims to describe the pathogenesis of S. aureus with respect to CF and recent studies and their findings using genotypic methods, which identify factors contributing to the adaptation of S. aureus and its evolution in the airways of individuals with CF. The subject of the review is up-to-date, and there are currently many studies presenting Staphylococcus aureus, especially MRSA, in the context of cystic fibrosis.

The authors divided their work into two parts, which makes the work readable and transparent. In the first part, they provide a very comprehensive overview of the pathogenesis of S. aureus. In the second part of the review, they focus on genotyping S. aureus in cystic fibrosis. The strength of the work is the extensive table which, in clear way, presenting the recent studies using sequencing approaches.

Although the subject of the work is interesting and overall the review is well written the manuscript has many errors which are listed below.

Minor specific suggestion/comments:

Minor specific suggestion/comments:

Line 4: “as a consequence” à “therefore”

Line 28: “S. aureus has to à “S. aureus must”

Line 54: “as a consequence” à “therefore/consequently”

Line 69: “so called” à “so-called”

Line 89: “which in particular arise” à “which arise”

Line 114: “colonize and infect humans” à “colonize, and infect humans”

Line 139: “turn” à “turns”

Line 164: “toxins [51] .” à “toxins [51].”

Line 193: “HGT has to” à “HGT must”

Line 203: “particular pathogenetic” à “pathogenetic”

Line 202: “aureus(CHIPS)” à “aureus (CHIPS)”

Line 221: “In accordance to” à “In accordance with”

Line 241: “are capable of causing” à “can cause”

Line 248: “In an attempt to” à “to”

Line 276: “but, particularly” à “but particularly”

Line 279: “In this context it also has to” à “In this context it also must”

Line 295: “a large number of” à “many”

Line 299: “in light of” à “considering”

Line 300: “most commonly used” à “most used”

Line 308: “analysed” à “analyzed”

Line 368: “A large number of” à “many”

Line 400: “et al.[52].” à “et al. [52].”

Line 441: “with regard to” à “regarding”

Line 443: “is able to” à “can”

Line 446: “with regard to” à “regarding”

Line 452: “two component” à “two-component”

Line 546: “recently, since” à “recently since”

Line 550: “was” à “were”

Line 656: “a number of” à “several”

Line 675: “in order to” à “to”

The summary of the work is an integral part of part 2. I would suggest creating a separate chapter „conclusion” from it.

Author Response

Reviewer 1

Comments and Suggestions for Authors

The article by Christine Rumpf et al. entitled “Staphylococcus aureus and cystic fibrosis – a close relationship. What can we learn from sequencing studies?” is a review article that aims to describe the pathogenesis of S. aureus with respect to CF and recent studies and their findings using genotypic methods, which identify factors contributing to the adaptation of S. aureus and its evolution in the airways of individuals with CF. The subject of the review is up-to-date, and there are currently many studies presenting Staphylococcus aureus, especially MRSA, in the context of cystic fibrosis.

The authors divided their work into two parts, which makes the work readable and transparent. In the first part, they provide a very comprehensive overview of the pathogenesis of S. aureus. In the second part of the review, they focus on genotyping S. aureus in cystic fibrosis. The strength of the work is the extensive table which, in clear way, presenting the recent studies using sequencing approaches.

Although the subject of the work is interesting and overall the review is well written the manuscript has many errors which are listed below.

Authors’ comment:

We would like to thank the reviewer for the positive evaluation of our review. The errors, which were mentioned by this reviewer, have been corrected accordingly.

Minor specific suggestion/comments:

Minor specific suggestion/comments:

Reviewer’s comment 1:

Line 4: “as a consequence” à “therefore”.

Revised manuscript line 4, has been corrected

Reviewer’s comment 2:

Line 28: “S. aureus has to à “S. aureus must”

Revised manuscript line 30, has been corrected

Reviewer’s comment 3:

Line 54: “as a consequence” à “therefore/consequently”

Revised manuscript line 63, has been corrected

Reviewer’s comment 4:

Line 69: “so called” à “so-called”

Revised manuscript line 78, has been corrected

Reviewer’s comment 5:

Line 89: “which in particular arise” à “which arise”

Revised manuscript line 102, has been corrected

Reviewer’s comment 6:

Line 114: “colonize and infect humans” à “colonize, and infect humans”

Revised manuscript line 129, has been corrected

Reviewer’s comment 7:

Line 139: “turn” à “turns”

Revised manuscript line 160, has been corrected to “which in turn reacts…”

Reviewer’s comment 8:

Line 164: “toxins [51] .” à “toxins [51].”

Revised manuscript line 188, has been corrected

Reviewer’s comment 9:

Line 193: “HGT has to” à “HGT must”

Revised manuscript line 218, has been corrected

Reviewer’s comment 10:

Line 203: “particular pathogenetic” à “pathogenetic”

Revised manuscript line 228, has been corrected

Reviewer’s comment 11:

Line 202: “aureus(CHIPS)” à “aureus (CHIPS)”

Revised manuscript lines 226/227, has been corrected

Reviewer’s comment 12:

Line 221: “In accordance to” à “In accordance with”

Revised manuscript line 249, has been corrected

Reviewer’s comment 13:

Line 241: “are capable of causing” à “can cause”

Revised manuscript line 269, has been corrected

Reviewer’s comment 14:

Line 248: “In an attempt to” à “to”

Revised manuscript line 279, has been corrected

Reviewer’s comment 15:

Line 276: “but, particularly” à “but particularly”

Revised manuscript line 306, has been corrected

Reviewer’s comment 16:

Line 279: “In this context it also has to” à “In this context it also must”

Revised manuscript line 309, has been corrected

Reviewer’s comment 17:

Line 295: “a large number of” à “many”

Revised manuscript line 326, has been corrected

Reviewer’s comment 18:

Line 299: “in light of” à “considering”

Revised manuscript line 336, has been corrected

Reviewer’s comment 19:

Line 300: “most commonly used” à “most used”

Revised manuscript line 336, has been corrected

Reviewer’s comment 20:

Line 308: “analysed” à “analyzed”

Revised manuscript line 344, has been corrected

Reviewer’s comment 21:

Line 368: “A large number of” à “many”

Revised manuscript line 411, has been corrected

Reviewer’s comment 22:

Line 400: “et al.[52].” à “et al. [52].”

Revised manuscript line 443, has been corrected

Reviewer’s comment 23:

Line 441: “with regard to” à “regarding”

Revised manuscript line 483, has been corrected

Reviewer’s comment 24:

Line 443: “is able to” à “can”

Revised manuscript line 485, has been corrected

Reviewer’s comment 25:

Line 446: “with regard to” à “regarding”

Revised manuscript line 488, has been corrected

Reviewer’s comment 26:

Line 452: “two component” à “two-component”

Revised manuscript line 493/494, has been corrected

Reviewer’s comment 27:

Line 546: “recently, since” à “recently since”

Revised manuscript line 595, has been corrected

Reviewer’s comment 28:

Line 550: “was” à “were”

Revised manuscript line 599, has been corrected

Reviewer’s comment 29:

Line 656: “a number of” à “several”

Revised manuscript line 707, has been corrected

Reviewer’s comment 30:

Line 675: “in order to” à “to”

Revised manuscript line 731, has been corrected

Reviewer’s comment 31:

The summary of the work is an integral part of part 2. I would suggest creating a separate chapter „conclusion” from it.

Revised manuscript: As this reviewer suggested, we created now a chapter “summary and conclusions”, lines 706ff.

Reviewer 2 Report

This manuscript is a review of use of bacterial sequencing in Staph aureus infection among people with cystic fibrosis.  Overall, this is a review that highlights basic concerns related to Staph infection in CF and focuses on why understanding the genetics of Staph could impact CF care in the future. Specifics of methods of sequencing as well as key genes thought to play a role in virulence and disease are reviewed.

Abstract: The flow and clarity of the abstract could be improved. A few specific questions include: Why is the abbreviation for single nucleotide variation defined as SNPs in the abstract.  SNP usually refers to single nucleotide polymorphisms.  Same applies to use of single or multi locus sequencing (spa-typing or MLST).  Make abbreviation align with the correct definition and place abbreviation in parenthesis immediately after the word(s). 

Body: Multiple areas require specific referencing. Lines 17-20 require a reference for prevalence of MSSA in the UK and use of continuous anti-staph treatment.  Line 25 has a reference missing.  Line 40 change genetical to genetic. Line 97 needs a reference cited. Need reference documented for sentence on line 390.

Figure specificity needs to be addressed: Figure 4: Figure 2 does not demonstrate how specific milieu conditions mediate regulatory mechanisms, it summarizes gene expression. It is a good figure but the intended description does not align with what is shown in the figure.  MGE is not clearly defined in the figure description. Figure 4 does not highlight the variation in MGEs.  This figure could be improved to show what is defined in the description. In figure 5 define virulence factors and adhesion genes, not clear which is which in the figure.

Fluidity of grammar could be improved.  Remove dash from genotypic- in line 57. Define "morphotypes" line 62.  Line 94 change rises to raises.  Line 116-118 poorly written and hard to follow. Line 120 change genetical to genetic. Line 234 unclear what "Dawkinse sense" refers to. Also a confusing sentence. whole genome sequencing mentioned in line 289, but later abbreviated in line 296. Line 317 use WGS (already defined). Line 377 define CHIPs.  Lines 661-662 rewrite sentence. Avoid sentence followed by: with another sentence.  Generally use a . to separate sentences.

Conclusions and future directions should be more clearly stated to highlight both results available to date and what specific future directions will impact better understanding of Staph infection in CF.

Excellent use of references from diverse sources. 

Author Response

Reviewer #2

Comments and Suggestions for Authors

This manuscript is a review of use of bacterial sequencing in Staph aureus infection among people with cystic fibrosis.  Overall, this is a review that highlights basic concerns related to Staph infection in CF and focuses on why understanding the genetics of Staph could impact CF care in the future. Specifics of methods of sequencing as well as key genes thought to play a role in virulence and disease are reviewed.

Authors’ comment: We would like to thank this reviewer for her/his appreciation of our review.

Reviewer’s comment:

Abstract: The flow and clarity of the abstract could be improved. A few specific questions include: Why is the abbreviation for single nucleotide variation defined as SNPs in the abstract.  SNP usually refers to single nucleotide polymorphisms.  Same applies to use of single or multi locus sequencing (spa-typing or MLST).  Make abbreviation align with the correct definition and place abbreviation in parenthesis immediately after the word(s). 

Authors’ comment: Reviewer 2 is right single nucleotide polymorphisms are abbreviated SNPs. The term single nucleotide variants (SNVs) is often used to describe SNPs, but to qualify as a SNP the variant must be present in at least 1% of the population. In our review we discuss both studies that look at SNVs as well as SNPs.

We changed the abbreviation from SNPs to SNVs.

Reviewer’s comment:

Body: Multiple areas require specific referencing. Lines 17-20 require a reference for prevalence of MSSA in the UK and use of continuous anti-staph treatment.  Line 25 has a reference missing. 

Authors’ comment: We thank the reviewer for her/his suggestion to include the respective references:

  • Ahmed MI, Mukherjee S. Treatment for chronic methicillin-sensitive Staphylococcus aureus pulmonary infection in people with cystic fibrosis. Cochrane Database Syst Rev 2018; 7:CD011581.
  • Goerke C, Wolz C. Adaptation of Staphylococcus aureus to the cystic fibrosis lung. Int J Med Microbiol 2010; 300:520-5.

Reviewer’s comment:

Line 40 change genetical to genetic.

Authors’ answer:

The word “genetical” was changed to “genetic” now in line 43.

Reviewer’s comment:

Line 97 needs a reference cited.

Authors’ answer:

We added the respective references:

  1. Foundation ACF. CFF Annual Patient Data Registry 2019. 2020.
  2. Society ECF. Annual data report (year 2018). 2020.

Reviewer’s comment:

Need reference documented for sentence on line 390.

Authors’ answer:

There are already 2 references mentioned at the end of the sentence.

  1. Westphal C, Gorlich D, Kampmeier S, et al. Antibiotic Treatment and Age Are Associated With Staphylococcus aureus Carriage Profiles During Persistence in the Airways of Cystic Fibrosis Patients. Front Microbiol 2020; 11:230.
  2. Ankrum A, Hall BG. Population Dynamics of Staphylococcus aureus in Cystic Fibrosis Patients To Determine Transmission Events by Use of Whole-Genome Sequencing. J Clin Microbiol 2017; 55:2143-52.

Therefore, we did not add another reference.

Reviewer’s comment:

Figure specificity needs to be addressed: Figure 2 does not demonstrate how specific milieu conditions mediate regulatory mechanisms, it summarizes gene expression. It is a good figure but the intended description does not align with what is shown in the figure.

Authors’ answer:

The reviewer is right, that the respective figure does not show the impact of the milieu conditions on the regulatory mechanisms. However, with this figure we intended to demonstrate the complexity of the underlying regulatory network: The phenotypical heterogeneity of S. aureus populations in CF patients is intensified by differences in gene expression, which is mediated by regulatory mechanisms responding to specific milieu conditions (Figure 2) [14].

Figure 2. S. aureus gene regulatory network. S. aureus regulatory genes, their reciprocal interference and their impact on adhesiveness (A) toxicity (T) and evasiveness (E) (adapted from Priest et al. (48)). Gene expression in S. aureus is regulated by a highly complex and not fully understood network of genes which mutually influence their activity (black arrow: induction, red arrow: inhibition).

Reviewer’s comment :

MGE is not clearly defined in the figure description. Figure 4 does not highlight the variation in MGEs. This figure could be improved to show what is defined in the description.

Authors’ answer:

In the revised version of the manuscript, the abbreviation of MGEs was added and the figure formatted accordingly. The figure is intended to show how much of the S. aureus genome belongs to the core and what proportion is variable between different strains.

Figure 4. S. aureus genome overview. 75% of S. aureus genes are found in more than 95% of the strains and represent the bacteria’s "core genome". The distribution of “core variable genes” differs between varying S. aureus strains. Mobile genetic elements (MGEs) such as bacteriophages, plasmids and pathogenicity islands, mainly encode for resistance proteins, superantigens and toxins. In addition to vertical transfer, they can be transmitted horizontally [45, 46].

Reviewer’s comment:

In figure 5 define virulence factors and adhesion genes, not clear which is which in the figure.

Authors’ answer:

To distinguish adhesion genes from toxin genes, we changed the format of the latter to bold italics.

Figure 5. S. aureus virulence genes (selected) and their localization. Virulence factors of S. aureus are localized on different sites. While many of the adhesin genes are located in the chromosome, toxins (in bold italics) are often located on plasmids, pathogenicity islands or carried and transduced by phages.

Reviewer’s comment:

Fluidity of grammar could be improved.

Remove dash from genotypic- in line 57.

Authors’ answer:

The dash was removed, now line 66.

Reviewer’s comment:

Define "morphotypes" line 62.

Authors’ answer:

Reviewer’s comment:

Line 94 change rises to raises.

Authors’ answer:

Correction was done, now line 107

Reviewer #2’s comment 14:

Line 116-118 poorly written and hard to follow.

Authors’ answer:

It is not clear, what the reviewer wants to point out with this criticsm. Therefore, we copied the respective lines in the rebuttal. These sentences describe the population structure of S. aureus and we can not follow the criticsm.

“S. aureus – fundamentals of pathogenicity

The population structure of S. aureus is clonal. Of all S. aureus strains, which have been isolated from humans, 10 dominant strains (referred to as "clonal complexes" (CC)), which can be subdivided into numerous sub-strains, colonize and infect humans [41, 42]. “

Reviewer’s comment:

Line 120 change genetical to genetic.

Authors’ answer:

Correction was done, now line 136.

Reviewer’s comment:

Line 234 unclear what "Dawkinse sense" refers to. Also a confusing sentence.

Authors’ answer:

We are sorry that this description was not clear enough. The evolutionary biologist Richard Dawkins described the theory of the gene-centered view of evolution in his book "The Selfish Gene" in 1976. This was cited many times and puts the adaptation and evolution of pathogens in hostile surroundings and survival of the fittest in a genomic context. “…in a Dawkins’ sense” refers to this.

We now changed the sentence, lines 261-264:

“It is presumed that HGT caused "neutral diversity" reflects the possibility that some MGEs represent "selfish DNA"  as pointed out by the evolutionary biologist Dawkins, who hypothesized that the spread of MGEs ly in their own interest, but not necessarily in the interest of the recipient [69].”

Reviewer’s comment:

whole genome sequencing mentioned in line 289, but later abbreviated in line 296.

Line 317 use WGS (already defined).

Authors’ answer:

Abbreviations have been reviewed and explained where they first appear.

Reviewer’s comment:

Line 377 define CHIPs.

Authors’ answer:

The abbreviation CHIPS was corrected and defined, line 227.

Reviewer’s comment:

Lines 661-662 rewrite sentence. Avoid sentence followed by: with another sentence.

Generally use a . to separate sentences.

Authors’ answer:

As the reviewer’s criticism was not clear to us, we are copying the sentence from the respective lines in this rebuttal letter.

“WGS of these isolates revealed several SNPs and mutations in various genes responsible for metabolism, virulence factors and regulators including agrA.”

In this sentence, we do not find the criticized sentence, which moved in the revised version of the manuscript to lines 711-713.

Reviewer’s comment:

Conclusions and future directions should be more clearly stated to highlight both results available to date and what specific future directions will impact better understanding of Staph infection in CF.

Authors’ answer:

As suggested by reviewer 1, we designated the last paragraph to “Summary and conclusions”. Also, we improved the organization of this last chapter, lines 714-758.

Reviewer’s comment:

Excellent use of references from diverse sources. 

Authors’ answer:

We would like to thank the reviewer for her/his appreciation.

Round 2

Reviewer 2 Report

This manuscript is a timely, relevant review.  The edits provided by the authors have addressed previously identified concerns and have substantially improved the presentation of the manuscript.  Thank you for your dedication to this excellent review. 

Author Response

Dear reviewer,

we would like to thank your for your thorough evaluation of our manuscript and the appreciation of the revised version of our manuscript.

with kind regards

Barbara Kahl